# State of lifestyle medicine education in Saudi medical schools: A descriptive study

**Mohammed Almansour[1], Abeer Salman Alzaben[2], Sadeem Abdulaziz Aljammaz[3], Hayat Saleh Alzahrani[4]***

1 Medical Education Department, College of Medicine, King Saud University, Riyadh, Saudi Arabia, 2 Department of Health Sciences, College of Health and Rehabilitation Sciences, Princess Nourah Bint Abdulrahman University, Riyadh, Saudi Arabia, 3 Community Health Sciences Department, College of Applied Medical Sciences, King Saud University, Riyadh, Saudi Arabia, 4 Department of Family and Community Medicine, Assistant Professor and Consultant of Family Medicine and Medical Education, College of Medicine, Princess Nourah bint Abdulrahman University, Riyadh, Saudi Arabia

* hsaalzahrani@pnu.edu.sa

## Abstract

### Background

Lifestyle medicine (LM) is a patient-centric, evidence-based clinical practice supporting adopting and sustaining of healthy behaviours and improving health-related living standards. Unfortunately, even in developed nations, medical curricula have largely ignored the LM concepts. Some LM components have been incorporated into the medical curriculum in the Kingdom of Saudi Arabia to determine the general status of teaching LM competency domains in undergraduate curricula in medical colleges.

### Methods

A cross-sectional, descriptive survey was conducted in English and distributed from January to March 2022. In this study, only administrative position holders were included. The administrative directors (such as deans, vice-deans, and program directors) play a significant role and are responsible for policymaking in medical education. There are 38 undergraduate medical programs across private and public medical colleges in Saudi Arabia. MCQs, OSCE, Essay, SEQ/SAQ, Assignments, and OSPE, were the assessment techniques of the LM domains.

### Results

The response rate of the survey was 78.3%. Of all respondents, 61% were aware of LM domains. Twenty-four colleges teach one or more of the LM domains; the nutrition domain was the most common one. Mostly followed the traditional method (75%) and small group learning activities (71%). Programs also used large group learning activities and clinical teaching (35% each), followed by practical laboratory activities (19%) and other methods on very few occasions.

**Data Availability Statement:** The datasets generated during and/or analysed during the current study are present within the study.

**Funding:** The author(s) received no specific funding for this work.

**Competing interests:** The authors have declared that no competing interests exist.

## Conclusion

The current study also shows that LM is not taught effectively in medical schools in Saudi Arabia, although the results illustrate an increased interest and awareness among administrators. This study identified the general situation of teaching LM in medical schools. These findings provide valuable insights for shaping the future direction of medical education.

## Introduction

Non-communicable diseases (NCDs) pose serious economic and health threats. Saudi Arabia is in the early phase of a demographic transition, and it is crucial to recognize the increasing NCD trends among its ageing population [1]. A recent microsimulation study predicted that direct healthcare costs of obesity-attributable diseases among the working-age population in Saudi Arabia would be over 127 billion USD by 2040 [2]. Clinicians, researchers, universities, and professional societies stress the significance of the association between unhealthy lifestyle behaviors and NCDs [3]. Modifiable behavioral risk factors influence the mortality rates [4]. The 2019 Global Burden of Disease study reports that improper dietary habits cause illness and death [5]. Thus, tackling lifestyle-related issues is the prominent treatment option for several chronic conditions [4].

Lifestyle medicine (LM) is a patient-centric, evidence-based clinical practice supporting healthy behaviors and improving health-related living standards [3]. LM is recognized as one of the five emerging medical domains in recent years, although it has documented proof of practice over two and a half millennia [4,6]. It uses therapeutic lifestyle interventions as a key treatment strategy. It involves the application of whole-person, prescriptive lifestyle changes to treat or even reverse chronic clinical conditions. The American College of LM identifies six key LM pillars: nutrition, physical activity, stress management, restorative sleep, social connection, and avoidance of risky substances [7]. The British Society of LM advocates three key principles: recognizing the need to work on socioeconomic health issues, implementing evidence-based approaches toward lasting positive paradigm shifts, and gaining knowledge on the importance of LM pillars, with an insight that none exists in seclusion [8].

Unfortunately, the LM concepts have largely been ignored in medical curricula, even in developed nations [9]. In a US-based study, third-year medical students (n = 115) appreciated the value of LM in the curriculum (100%) and showed a willingness to learn (98%). However, they were not confident enough to set life-changing goals for patients (3.1±0.9) [10]. In another US study, 84% of clinical students (n = 74) reportedly spent less time on LM aspects. Among them, only 56%, 54%, 58%, and 25% had the required skill to perform a physical examination for authorizing an exercise program, ascertaining maximal heart rate, developing a nutritional plan, and planning an exercise prescription, respectively [11]. Several recent research works signify the importance of incorporating LM curricula in medical education [12–14]. Trilk et al. provided insights on developing an LM-based curriculum in 2012 at the University of South Carolina School of Medicine Greenville, USA [3]. They used an evidence-supported four-step instructional design (i.e., Analysis, Design, Development, and Evaluation) for formulating and implementing the LM curriculum, which focuses on delivering total healthcare [3]. There is an increasing acceptance of LM concepts among medical students, and the lessons help shape their behaviors [10,11,14].

There are 36 medical colleges (public: 29, private: 7) in Saudi Arabia. Some LM components have been incorporated into the medical curriculum in the Kingdom. There is a shortage of

systematic implementation of all LM components. Moreover, Saudi Commission for Health Specialties offers an LM fellowship program [15]. The Saudi Vision 2030, a policy implementation that aims to bring substantial changes in all sectors, including healthcare, vouches for the significance of improving overall health by curtailing NCDs. This offers an ideal chance to work on modernizing the curricula of medical schools and incorporating all LM elements into them. Limited studies have assessed all LM components among medical students globally, and no studies have assessed the teaching of LM components in Saudi medical colleges. Thus, the current study aimed to determine the general status of teaching LM competency domains in undergraduate curricula in Saudi medical colleges.

## Materials and methods

### Participation of the study

A cross-sectional, descriptive survey was conducted in English and distributed from 28 January to 31 March 2022. In this study, only administrative position holders were invited and recruited. The Administrative Directors (such as Deans, Vice-Deans, and Program Directors) play a significant role and hold major responsibilities in policymaking within medical education. There are 38 undergraduate medical programs across private and public medical colleges in Saudi Arabia. The study aimed to involve at least one administrative position holder from each of these 38 medical colleges to gain an accurate understanding of the current curriculum. This information is crucial for guiding the future direction of medical colleges. The medical college program director manages many responsibilities, one of the most important being the curriculum, evaluation, and supervision of residents and fellows, including clinical and educational aspects. This study is crucial for understanding medical directors' current perspectives and medical colleges' future direction. The survey was developed using the REDCap (Research Electronic Data Capture) application hosted at King Saud University, Riyadh, Saudi Arabia. The survey link (https://rcmed.ksu.edu.sa/surveys/?s=8PM4HJDCD8) was shared via email. Study data were collected and managed using REDCap electronic data capture tools hosted at King Saud University. All participants were fully informed about the study's objectives and were entitled to skip any questions they did not wish to answer.

### Data collection

A simple, detailed, and relevant self-administered questionnaire in English was developed using scholarly sources and an exhaustive literature review to fulfil the study's objectives [3,16,17]. The questionnaire and objectives were thoroughly discussed with a panel of ethical team members and two medical education experts. They suggested validating the questions before conducting the final study and sharing the preliminary results with the moral and medical education teams. A reliability pre-test with 20 participants was performed, and Cronbach's alpha coefficient of 0.789 was obtained, within the acceptable limit of $>0.7$. These results were shared with the teams. The final study proceeded after incorporating their suggestions and receiving clearance.

### Study setting

The survey was comprised of four parts: The first part covered the sociodemographic information of the participants (age, gender, experience, administrative position) alongside the general information regarding the undergraduate medical programs (admission qualification, gender of candidates). The second part was about teaching each LM domain in the curriculum. There were three options: yes (whole course), yes (part of a course), or no.

The third part covered the teaching status of each LM domain. The questions were open-ended, optional questions asking about years taught in each domain, the number of program learning outcomes and courses learning outcomes, credit hours of courses, and total contact hours.

The fourth part was about methods of teaching and evaluation. The possible responses for the teaching methods were lectures, small-group learning sessions (problem-based learning (PBL), Case-based discussions, discussion groups, etc.), large-group learning sessions (seminars, tutorials, etc.), and practical laboratory and clinical teaching (bedside teaching, grand rounds, clinical encounters, etc.). The possible responses for the evaluation methods were multiple choice questions (MCQs), short-answer questions (SAQs)/ short-essay questions (SEQs), essay questions, assignments, Objective Structured Clinical Examination (OSCE)/Objective Structured Practical Examination (OSPE), oral exam, PBL and Clinical Practice checklist. The participants were able to choose more than one answer for this part.

## Data analysis

The data were analyzed using SPSS software version 26. Descriptive data are presented as means and standard deviation for continuous variables and frequencies and (%) for categorical variables.

## Ethical approval

The ethical approval was obtained from the Institutional Review Board Committee of King Saud University (E-21-6373). Written informed consent was taken from all participants.

## Results

The study included a total of 34 participants. The response rate of the survey was 78.3% (29/37). The survey was answered by two representatives from five medical schools and one each from the remaining schools. The mean age of the participants was 45.4 ± 8.4 years. Regarding gender distribution, 22 participants (66.6%) were male and 11 (33.3%) were female. The respondents were senior academicians with an average of 15.8±10.2 years of professional experience. Two-thirds were male, and most (73.4%) had administrative assignments for over two years. Their academic titles included Vice Dean (32.3%), Head of Medical Education Department/Unit (29.4%), and Program Director (17.6%).

The types of curricula in the MBBS programs at different medical colleges varied: 2 participants (5.8%) were involved in traditional programs, 14 (41.1%) in PBL-based programs, 17 (50%) in hybrid programs, 11 (32.3%) in outcome-based programs, and 1 (2.9%) in TBL-based programs. The total duration of the MBBS programs also varied, with 26 participants (76.4%) indicating a duration of 6 years plus a 1-year internship and 8 participants (23.5%) indicating a duration of 5 years plus a 1-year internship (Table 1).

The majority of the representatives reported that their undergraduate medical programs enrol both genders (91.1%) and high school graduates (85.2%), and the study lasts for six years with a 1-year internship (76.4%). Half (50%) of those programs have hybrid-type curricula, while the rest have adopted PBL (41%), outcome-based (32.3%), and traditional (5.8%) curricula, and one college (2.9%) has adopted Team-based learning (TBL). Of all respondents, 61% were aware of LM domains. Five respondents indicated that they have a specific course/module/block to teach LM (Table 1).

Further analysis revealed that 24 colleges teach one or more of the LM domains. The nutrition domain is the most common one. Six colleges (17.6%) teach nutrition as a standalone course, and fourteen (41.1%) as part of a course. Physical activity and health and wellness coaching are taught as a standalone course in one college (2.9%) and as part of a course in

Table 1. Population characteristics and practice(n = 34).

| | N (%) or mean ± SD |
|---|---|
| **Age (mean ± SD)[1]** | **45.4 ± 8.4** |
| **Gender [2]** | |
| *Male* | 22 (66.6%) |
| *Female* | 11 (33.3%) |
| **Experience in academia (mean ± SD) [2]** | 15.8 ± 10.2 |
| **Number of years in administrative assignment** | |
| **<1** | 2 (5.8%) |
| **1–2** | 7 (20.5%) |
| **2–4** | 6 (17.6%) |
| **4–6** | 10 (29.4%) |
| **>6** | 9 (26.4%) |
| **Administrative position*** | |
| *Dean* | 2 (5.8%) |
| *Vice-dean* | 11 (32.3%) |
| *Program director* | 6 (17.6%) |
| *Head of medical education department/unit* | 10 (29.4%) |
| *Curriculum unit head* | 3 (8.8%) |
| *Chairman, Basic Science* | 1 (2.9%) |
| *Basic Phase consultant* | 1 (2.9%) |
| *Program/ course/ block coordinator Course coordinator* | 3 (8.8%) |
| *Head of Quality department* | 2 (5.8%) |
| *Head of Department* | 1 (2.9%) |
| **The type of the MBBS program** | |
| *Traditional* | 2 (5.8%) |
| *PBL based* | 14 (41.1%) |
| *Hybrid* | 17 (50%) |
| *Outcome based* | 11 (32.3%) |
| *TBL based* | 1 (2.9%) |
| **The total duration of your MBBS program (including preparatory year** | |
| *6 years and 1 year internship* | 26 (76.4%) |
| *5 years and 1 year internship* | 8 (23.5%) |
| **The admission qualification to your MBBS program** | |
| *High school graduates* | 29 (85.2%) |
| *Bachelor graduates* | 3 (8.78%) |
| *High school and Bachelor for stream 2* | 1 (2.9%) |
| *MD Degree* | 1 (2.9%) |
| **The gender of candidates admitted to your MBBS program** | |
| *Male only* | 2 (5.8%) |
| *Female only* | 1 (2.9%) |
| *Both* | 31 (91.1%) |
| **Awareness of the Lifestyle Medicine competencies** | |
| *Yes* | 21 (61.7%) |
| *No* | 3 (8.8%) |
| *I am not sure* | 10 (29.4%) |
| **Have a specific course/module/block to teach Lifestyle Medicine** | |
| *Yes* | 5 (14.7%) |
| *No* | 29 (85.2%) |

[1] n = 28.

[2] n = 33.

seven (20.5%) and five (14.7%) colleges, respectively. Other LM domains are taught as a part of a course in some colleges, while some colleges do not teach at all (Table 2).

The details of teaching courses on Lifestyle Medicine (LM) competencies are presented in Table 3. The courses varied in duration, credit hours, total contact hours, and the domains they covered.

**Table 2. LM competencies' domains teaching status.**

| | Specific course/module/block to teach any of these competencies as lifestyle interventions | | |
|---|---|---|---|
| Domains | Yes (whole course) | Yes (part of a course) | No |
| Alcohol Use | 0 | 11 (32.3%) | 23 (67.6%) |
| Emotional Wellness | 0 | 11 (32.3%) | 23 (67.6%) |
| Health and Wellness Coaching | 1 (2.9%) | 5 (14.7%) | 28 (82.3%) |
| Mindfulness | 0 | 6 (17.6%) | 28 (82.3%) |
| Nutrition | 6 (17.6%) | 14 (41.1%) | 14 (41.1%) |
| Physical Activity | 1 (2.9%) | 7 (20.5%) | 26 (76.4%) |
| Sleep Health | 0 | 5 (14.7%) | 29 (85.2%) |
| Tobacco Cessation | 0 | 15 (44.1%) | 19 (55.8%) |
| Weight Management | 0 | 14 (41.1%) | 20 (58.8%) |

In this study, alcohol Use was taught for an average of 5.3 ± 1 years, covering 1.5 ± 0.3 Program Learning Outcomes (PLOs) and 2.6 ± 0.5 Course Learning Outcomes (CLOs). The course lasted 6.8 ± 1.3 weeks with 6 ± 1.7 credit hours and 102.4 ± 25.1 total contact hours. It was a domain as part of a course in 11 colleges. Furthermore, Emotional Wellness had an average teaching duration of 9 ± 1.1 years, with 1.6 ± 0.3 PLOs and 4 ± 0.8 CLOs. The course duration was 4.5 ± 0.7 weeks, with 4.6 ± 1.1 credit hours and 77.7 ± 19.7 total contact hours. This competency was included as a domain as part of a course in 11 colleges.

Moreover, Health and Wellness Coaching was taught for 5.4 ± 1.2 years, with 1.8 ± 0.4 PLOs and 2.6 ± 0.6 CLOs. The course duration was 4.5 ± 1.6 weeks, with 3.6 ± 1.4 credit hours and 72.6 ± 31.9 total contact hours. It was a domain as part of a course in 5 colleges and a whole course in 1 college. A majority of colleges taught Nutrition for an average of 7.1 ± 1.02 years, with 3.6 ± 0.9 PLOs and 4.9 ± 1.03 CLOs. The course lasted 7.2 ± 1.01 weeks, with 5.2 ± 0.7 credit hours and 70.8 ± 16.8 total contact hours. It was a domain as part of a course in 14 colleges and as a whole course in 6 colleges. Physical activity had an average teaching duration of 5.6 ± 1.2 years, with 2.2 ± 0.3 PLOs and 3 ± 0.4 CLOs. The course duration was 4.7 ± 2.1 weeks, with 5.8 ± 2.2 credit hours and 105.4 ± 38.5 total contact hours. It was a domain as part of a course in 7 colleges and a whole course in 1 college (Table 3).

Lectures on LM mostly followed the traditional method (75%) and small group learning activities (71%). Programs also used large group learning activities and clinical teaching (35% each), followed by practical laboratory activities (19%) and other methods on very few occasions.

In the vertical analysis for each domain's teaching methods, physical activity follows the same pattern of overall frequency as above. At the same time, nutrition has the percentage of lectures, small-group learning, practical lab, large-group learning, clinical teaching, and others. Lectures and small-group learning shared the same percentage, followed by clinical teaching and large-group learning in health and wellness coaching. Interestingly, small-group learning was found to be more frequently used in teaching alcohol use control and mindfulness than others (Table 4).

MCQs, OSCE, Essay, SEQ/SAQ, Assignments, and OSPE were the assessment techniques of the LSM domains that corresponded with their teaching methodologies. Most domains have been evaluated using many methods, including written and practical examinations. The majority of the time, all three course-specific areas (nutrition, physical activity, and health & wellness coaching) were evaluated by written assessments (Table 4).

## Discussion

The study provided an opportunity to investigate the current practices of medical schools about LM. There was an excellent response rate, with results showing that most medical

Table 3. Details of teaching courses of LM competencies.

| | Years been taught* | PLOs * | CLOs* | Duration (weeks)* | Credit hours of courses * | Total contact hours * | Domain as part of course [†] | Domain as whole Course [†] |
|---|---|---|---|---|---|---|---|---|
| Alcohol Use | 5.3 ± 1 (n = 9) | 1.5 ± 0.3 (n = 9) | 2.6 ± 0.5 (n = 9) | 6.8 ± 1.3 (n = 9) | 6 ± 1.7 (n = 8) | 102.4 ± 25.1 (n = 7) | 11 | 0 |
| Emotional Wellness | 9 ± 1.1 (n = 9) | 1.6 ± 0.3 (n = 8) | 4± 0.8 (n = 9) | 4.5 ± 0.7 (n = 8) | 4.6 ± 1.1 (n = 9) | 77.7 ± 19.7 (n = 9) | 11 | 0 |
| Health and Wellness Coaching | 5.4 ± 1.2 (n = 5) | 1.8 ± 0.4 (n = 5) | 2.6 ± 0.6 (n = 5) | 4.5 ± 1.6 (n = 4) | 3.6 ± 1.4 (n = 5) | 72.6 ± 31.9 (n = 5) | 5 | 1 |
| Mindfulness | 6.4 ± 1.2 (n = 5) | 1.8 ± 0.5 (n = 5) | 3 ± 1.3 (n = 4) | 3.7± 1.03 (n = 4) | 4.7 ± 1.7 (n = 4) | 80.5 ± 34.9 (n = 4) | 6 | 0 |
| Nutrition | 7.1 ± 1.02 (n = 16) | 3.6 ± 0.9 (n = 15) | 4.9 ± 1.03 (n = 15) | 7.2 ± 1.01 (n = 15) | 5.2 ± 0.7 (n = 14) | 70.8 ± 16.8 (n = 15) | 14 | 6 |
| Physical Activity | 5.6 ± 1.2 (n = 5) | 2.2 ± 0.3 (n = 5) | 3 ± 0.4 (n = 5) | 4.7 ± 2.1 (n = 4) | 5.8 ± 2.2 (n = 5) | 105.4 ± 38.5 (n = 5) | 7 | 1 |
| Sleep Health | 7 ± 2.2 (n = 4) | 4.7 ± 3.1 (n = 4) | 5.2 ± 2.9 (n = 4) | 4.5 ± 1.6 (n = 4) | 6.5 ± 1.5 (n = 4) | 133.5 ± 27.8 (n = 4) | 5 | 0 |
| Tobacco Cessation | 5.7 ± 1.2 (n = 10) | 1.3 ± 0.3 (n = 10) | 1.9 ± 0.3 (n = 10) | 6.7 ± 1.4 (n = 9) | 5.2 ± 1.5 (n = 10) | 57.7 ± 23.3 (n = 9) | 15 | 0 |
| Weight Management | 7.1 ± 1.2 (n = 11) | 3.4 ± 1.2 (n = 10) | 3.8 ± 1.2 (n = 10) | 6.3 ± 1.2 (n = 9) | 6.6 ± 1.4 (n = 10) | 101 ± 22.4 (n = 9) | 14 | 0 |

*mean ± standard error.

† Number of colleges.

schools in Saudi Arabia were aware of the importance of teaching LM to medical students. Many colleges have been teaching one or more of the LM domains for years. Of the 29 medical schools that completed the survey, 24 schools taught one or more of the LM Domains, and 15 taught three or more LM components as a standalone course or mostly as parts of courses. This is on par with the worldwide direction for integrating LM early in medical training and practice [18–22].

Nonetheless, medical schools in Saudi Arabia seem to have a comparable LM integration to many international medical schools, though not all components are introduced in medical programs. LM researchers acknowledge that despite the wealth of evidence through the past two decades recognizing LM's remarkable positive health and economic outcomes and a demand by students and practitioners to increase training, the medical community is still slow to respond to changes [16,23,24].

On average, 5 credits of LM domains are added as parts of medical courses. This may indicate that they provide introductory information with reduced emphasis on competence for students in these domains, as seen at many medical schools worldwide [5–15]. However, six medical schools teach standalone courses dedicated to nutrition, and one school teaches health and wellness coaching and physical activity as standalone courses. Surprisingly, only one medical school has a dedicated course for physical activity, and only seven teach it as part of a course, even though there is a direction for physical activity promotion in Saudi Arabia and a consensus statement from the Saudi Public Health Authority to include physical activity in medical care [25,26]. However, it is also encouraging that these medical schools have recognized the importance of providing a standalone course for each domain. Researchers have found that medical students' knowledge of nutrition and physical activity is generally inadequate, even though these domains have been proven to be essential for health for a long time [27–29].

However, the finding that teaching only one course per domain is sufficient in providing the medical student with competency in these domains will warrant further investigation.

**Table 4. Instructional and assessment methods of LM competencies.**

| | Alcohol Use | Emotional Wellness | Health and Wellness Coaching | Mindfulness | Nutrition | Physical Activity | Sleep Health | Tobacco Cessation | Weight Management | Overall frequency |
|---|---|---|---|---|---|---|---|---|---|---|
| **What are the instructional methods used to teach it? (multiple answers allowed)** | | | | | | | | | | |
| Lectures | 9 (26.4%) | 9 (26.4%) | 4 (11.7%) | 5 (14.7%) | 17 (50%) | 7 (20.5%) | 4 (11.7%) | 9 (26.4%) | 11 (32.3%) | 75 |
| Small group learning sessions (PBL, CBD, Discussion group...etc) | 10 (29.4%) | 9 (26.4%) | 4 (11.7%) | 6 (17.6%) | 14 (41.1%) | 6 (17.6%) | 3 (8.8%) | 9 (26.4%) | 10 (29.4%) | 71 |
| Large group learning sessions (seminars, tutorial...etc) | 4 (11.7%) | 5 (14.7%) | 2 (5.8%) | 2 (5.8%) | 6 (17.6%) | 3 (8.8%) | 3 (8.8%) | 4 (11.7%) | 6 (17.6%) | 35 |
| Practical lab | 2 (5.8%) | 1 (2.9%) | 1 (2.9%) | 1 (2.9%) | 8 (23.5%) | 2 (5.8%) | 1 (2.9%) | 0 | 3 (8.8%) | 19 |
| Clinical teaching (bedside teaching, grand rounds, clinical encounters.. etc) | 3 (8.8%) | 6 (17.6%) | 3 (8.8%) | 4 (11.7%) | 4 (11.7%) | 2 (5.8%) | 3 (8.8%) | 4 (11.7%) | 6 (17.6%) | 35 |
| Field visits | 1 (2.9%) | 0 | 1 (2.9%) | 0 | 2 (5.8%) | 0 | 0 | 0 | 2 (5.8%) | 6 |
| PCC | 1 (2.9%) | 0 | 0 | 0 | 0 | 0 | 0 | 0 | 0 | 1 |
| Case Presentation | 0 | 1 (2.9%) | 0 | 0 | 0 | 0 | 0 | 0 | 0 | 1 |
| Assignment | 0 | 0 | 0 | 0 | 1 (2.9%) | 0 | 0 | 0 | 0 | 1 |
| **What are the assessment methods used to evaluate students? (multiple answers allowed)** | | | | | | | | | | |
| MCQs | 10 (29.4%) | 10 (29.4%) | 5 (14.7%) | 5 (14.7%) | 19 (55.8%) | 7 (20.5%) | 4 (11.7%) | 12 (35.2%) | 12 (35.2%) | 84 |
| SEQ/SAQ | 4 (11.7%) | 7 (20.5%) | 2 (5.8%) | 6 (17.6%) | 10 (29.4%) | 5 (14.7%) | 2 (5.8%) | 4 (11.7%) | 8 (23.5%) | 48 |
| Essay questions | 2 (5.8%) | 0 | 1 (2.9%) | 1 (2.9%) | 3 (8.8%) | 2 (5.8%) | 1 (2.9%) | 1 (2.9%) | 3 (8.8%) | 59 |
| Assignment | 4 (11.7%) | 3 (8.8%) | 3 (8.8%) | 4 (11.7%) | 7 (20.5%) | 4 (11.7%) | 2 (5.8%) | 2 (5.8%) | 2 (5.8%) | 34 |
| OSCE | 3 (8.8%) | 4 (11.7%) | 3 (8.8%) | 2 (5.8%) | 6 (17.6%) | 3 (8.8%) | 4 (11.7%) | 4 (11.7%) | 7 (20.5%) | 65 |
| OSPE | 3 (8.8%) | 1 (2.9%) | 1 (2.9%) | 0 | 7 (20.5%) | 1 (2.9%) | 1 (2.9%) | 2 (5.8%) | 2 (5.8%) | 18 |
| Oral exam | 0 | 1 (2.9%) | 0 | 0 | 0 | 0 | 1 (2.9%) | 1 (2.9%) | 1 (2.9%) | 4 |
| **PBL and CP checklist** | 0 | 1 (2.9%) | 0 | 0 | 0 | 0 | 0 | 0 | 0 | 1 |

Successful examples of incorporating LM curricula at several medical schools in the United States suggest that more teaching hours are needed to reach competency [13,30–32].

The study results showed that nearly 40% of the participants were unaware of LM competencies. On average, 60% of the schools did not teach all LM competencies, raising the point of how efficiently students learn about chronic illnesses and preventative medicine. Chronic illnesses are the cause of substantial suffering, and it has been proven that lifestyle changes are an integral part of preventing and treating chronic illnesses [17–19]. Therefore, the recommendation to include LM components in medical school curricula and training is to have it as a competency for all medical specialities, regardless of whether they decide to pursue it as a speciality, with the development and implementation of policies to help foster the needed curricula changes [21–23]. Some researchers go beyond medical school curricula to recommend designing and implementing a new healthcare model that efficiently incorporates LM in the health systems [17,33].

The LM domains taught in Saudi medical schools utilize several blended teaching methods but mostly rely on lectures and small-group learning. Blended learning strategies are preferred by students, as these differ in learning styles and provide knowledge. There is a noticeable universal deviation from traditional instructional methods to other exploratory methods, such as

simulation-based learning, reflection, small-group learning activities, and eLearning, which should improve learning [34,35]. Medical schools in Saudi Arabia might benefit from introducing more blended teaching strategies, including team-based learning, case presentations, and simulation, which would, in turn, benefit from incorporating LM teaching within the fabric of medical school curricula [36,37].

Assessment methods used at the different Saudi medical schools are similar to most of what is practised internationally; that is, most of the assessment methods are designed to assess knowledge competencies, and clinical assessments assess practical and clinical skills [38]. Assessments highly impact the quality of learning. Ensuring proper assessment is paramount in medical education, and it must go beyond only assessing medical expertise and include assessing communication skills, leadership, and professionalism [38]. Several researchers have identified and recommended the best practices for assessment; some of the main recommendations are to include self-assessment practices by medical students, use triangulation by including different assessment methods and have clearly defined expectations and competencies that will be assessed [38]. These recommendations are best incorporated within any update of medical school curricula during the addition of LM teaching.

This provides the perfect opportunity to update the medical curricula and incorporate LM components, which echoes Saudi Arabia's "Vision 2030" New Model of Patient-Centric Care [39,40].

There are some limitations to the current study. The data was collected through an online survey. The study relied on administrators' self-reports; the researchers did not study the curricula content at each medical school. In addition, one section of the survey was optional, which reduced the amount of information received. Furthermore, the current study is purely descriptive and doesn't include regression analysis. Future studies should be conducted on a broader scale and include regression analysis.

## Conclusion

The study highlighted the diversity in MBBS program types, with traditional, PBL-based, hybrid, outcome-based, and TBL-based curricula represented. The study also underscored the variety of instructional and assessment methods used to teach Lifestyle Medicine (LM) competencies, reflecting the multifaceted approach required to train medical students effectively. The current study also shows that LM is not taught effectively in medical schools in Saudi Arabia, although the results illustrate an increased interest and awareness among administrators. This study identified the general situation of teaching LM in medical schools. These findings provide valuable insights for shaping the future direction of medical education. However, further studies are needed to evaluate the LM teaching situation systematically.

## Recommendation

The first recommendation is to share these data with the Dean's Committee of Saudi Medical Schools to incorporate LSM into medical curricula in Saudi Arabia. Second, education grants can be utilized to stimulate the process. Finally, it is necessary to comprehend the obstacles surrounding implementing and incorporating LM into medical school curricula.

## Acknowledgments

Princess Nourah bint Abdulrahman University Researchers Supporting Project number (PNURSP2024R354), Princess Nourah bint Abdulrahman University, Riyadh, Saudi Arabia.

## Author Contributions

**Conceptualization:** Mohammed Almansour, Hayat Saleh Alzahrani.

**Data curation:** Mohammed Almansour, Abeer Salman Alzaben, Sadeem Abdulaziz Aljammaz.

**Formal analysis:** Sadeem Abdulaziz Aljammaz.

**Methodology:** Abeer Salman Alzaben.

**Supervision:** Hayat Saleh Alzahrani.

**Writing – original draft:** Abeer Salman Alzaben, Sadeem Abdulaziz Aljammaz.

**Writing – review & editing:** Mohammed Almansour, Hayat Saleh Alzahrani.

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
