## [Decision Letter · Decision Letter 0]

14 Jun 2024

PONE-D-24-21325State of lifestyle medicine education in Saudi medical schoolsPLOS ONE

Dear Dr. Alzahrani,

Thank you for submitting your manuscript to PLOS ONE. After careful consideration, we feel that it has merit but does not fully meet PLOS ONE’s publication criteria as it currently stands. Therefore, we invite you to submit a revised version of the manuscript that addresses the points raised during the review process. Please submit your revised manuscript by Jul 29 2024 11:59PM. If you will need more time than this to complete your revisions, please reply to this message or contact the journal office at plosone@plos.org. Please include the following items when submitting your revised manuscript:A rebuttal letter that responds to each point raised by the academic editor and reviewer(s). You should upload this letter as a separate file labeled 'Response to Reviewers'.A marked-up copy of your manuscript that highlights changes made to the original version. You should upload this as a separate file labeled 'Revised Manuscript with Track Changes'.An unmarked version of your revised paper without tracked changes. You should upload this as a separate file labeled 'Manuscript'.

We look forward to receiving your revised manuscript.

Kind regards,

Mukhtiar Baig, Ph.D.

Academic Editor

PLOS ONE

Journal Requirements:

2. In the online submission form, you indicated that [The datasets generated during and/or analysed during the current study are available from the corresponding author upon reasonable request.]. 

Reviewers' comments:

Reviewer's Responses to Questions

**Comments to the Author**

1. Is the manuscript technically sound, and do the data support the conclusions?

Reviewer #1: Yes

Reviewer #2: No

2. Has the statistical analysis been performed appropriately and rigorously? 

Reviewer #1: Yes

Reviewer #2: No

3. Have the authors made all data underlying the findings in their manuscript fully available?

Reviewer #1: Yes

Reviewer #2: No

4. Is the manuscript presented in an intelligible fashion and written in standard English?

Reviewer #1: Yes

Reviewer #2: No

5. Review Comments to the Author

Reviewer #1: 1. Abstract:

1.1. In abstract, method section, the statement: (We aimed

to involve at least one administrative position holder from each of these 38 medicals

colleges in this study to gain an accurate understanding of the current curriculum.) is not clear.

1.2. The result section in abstract is deficient, it is not supported by enough findings.

1.3. The authors mentioned in result section that (All three course specific areas (nutrition, physical activity, and health & wellness coaching) were mostly evaluated by written assessments) This finding is not supported by number or percentage.

1.4. In abstract, result section, the authors wrote (MCQs, OSCE, Essay, SEQ/SAQ, Assignments, and OSPE were the assessment techniques of the LM domains). This statement is belong to methodology section not a result.

2. Introduction: suitable and expressive.

The objectives are clear, and so the presentation

3. Presentation and methods: the methods are well described.

4. Results section:

This part is good but I suggest the following:

4.1. The author should add a regression table to demonstrate the factors that influence the deficiency in awareness of LM among the participants.

Reviewer #2: The English language needs to be revised by either a professional English editing agency or a native speaker.

Avoid using pronouns such as "we".

The statistics lacks any sign for bivariate analysis.

No limitations stated or future studies.

Your conclusion can be shortened and focus on the outcomes.

6. PLOS authors have the option to publish the peer review history of their article (what does this mean?). If published, this will include your full peer review and any attached files.

Reviewer #1: **Yes: **Moawia Bushra Gameraddin

Reviewer #2: No

---

## [Author Response · Author response to Decision Letter 0]

11 Jul 2024

Editorial comments 

In the online submission form, you indicated that [The datasets generated during and/or analysed during the current study are available from the corresponding author upon reasonable request.]. 

 The data generated for the current study is already present in the study in terms of frequency and percentage. Thank you for pointing out our mistake, we have also corrected the statement in the manuscript. 

Your ethics statement should only appear in the Methods section of your manuscript. If your ethics statement is written in any section besides the Methods, please move it to the Methods section and delete it from any other section. Please ensure that your ethics statement is included in your manuscript, as the ethics statement entered into the online submission form will not be published alongside your manuscript. 

We have made the changes as per the journal style and now ethical statement is in the method section.

Reviewer 1 Thank you for your thorough review and valuable feedback. We have made several revisions to address your comments and improve the manuscript. Below are our responses to each of your points:

Reviewer #1: 1. Abstract:

1.1. In abstract, method section, the statement: (We aimed to involve at least one administrative position holder from each of these 38 medicals

colleges in this study to gain an accurate understanding of the current curriculum.) is not clear. We have made the suggested changes. 

1.2. The result section in abstract is deficient, it is not supported by enough findings. Thank you for the suggestion. We have made the specified changes in abstract result section. 

1.3. The authors mentioned in result section that (All three course specific areas (nutrition, physical activity, and health & wellness coaching) were mostly evaluated by written assessments) This finding is not supported by number or percentage. We have made the specified changes.

1.4. In abstract, result section, the authors wrote (MCQs, OSCE, Essay, SEQ/SAQ, Assignments, and OSPE were the assessment techniques of the LM domains). This statement is belong to methodology section not a result. Thank you for pointing this out. The authors appreciate this comment. It is added in the conclusion in the revised version of the manuscript

2. Introduction: suitable and expressive.

The objectives are clear, and so the presentation We appreciate your positive feedback on the introduction. We are glad that you found it suitable and expressive.

3. Presentation and methods: the methods are well described.

Thank you for acknowledging that the methods are well described. We have ensured that they are detailed and clear to facilitate replication and understanding.

4. Results section:

This part is good but I suggest the following:

4.1. The author should add a regression table to demonstrate the factors that influence the deficiency in awareness of LM among the participants.

 Our analysis was designed to be simple and descriptive, as the primary objective of our study was to explore the general status of Lifestyle Medicine (LM) teaching across medical colleges in Saudi Arabia. We aimed to provide an overview rather than conducting detailed comparisons of the characteristics of each college.

- Given this objective, we have not included a regression table in the results section. Instead, we focused on presenting descriptive statistics to provide a comprehensive picture of the current state of LM teaching.

Reviewer #2: Thank you for your thorough review and valuable feedback. We have made several revisions to address your comments and improve the manuscript. Below are our responses to each of your points:

The English language needs to be revised by either a professional English editing agency or a native speaker.

 We have revised the manuscript for clarity and readability. The manuscript has been reviewed by a professional English editing agency to ensure proper language usage and grammatical accuracy. The certificate of English editing has been uploaded as supplementary file.

Avoid using pronouns such as "we". The manuscript has been edited to avoid the use of pronouns such as "we." Statements have been rephrased to maintain an objective tone.

The statistics lacks any sign for bivariate analysis.

 Our study aimed to provide a descriptive overview of the current state of LM teaching rather than conducting detailed statistical comparisons. Therefore, bivariate analysis was not included in the original design. However, we have clarified this in the manuscript to ensure the study's scope and objectives are clear.

No limitations stated or future studies.

 A section on limitations and suggestions for future studies has been added to the manuscript.

Your conclusion can be shortened and focus on the outcomes. The conclusion has been shortened and now focuses more directly on the key outcomes of the study. The revised conclusion emphasizes the primary findings and their implications for the future direction of LM education in Saudi medical schools.

---

## [Editor Report · Decision Letter 1]

25 Jul 2024

State of lifestyle medicine education in Saudi medical schools: A descriptive study

PONE-D-24-21325R1

Dear Dr. Alzahrani,

We’re pleased to inform you that your manuscript has been judged scientifically suitable for publication and will be formally accepted for publication once it meets all outstanding technical requirements.

Kind regards,

Mukhtiar Baig, Ph.D.

Academic Editor

PLOS ONE

---

## [Editor Report · Acceptance letter]

31 Jul 2024

PONE-D-24-21325R1 

PLOS ONE

Dear Dr. Alzahrani, 

I'm pleased to inform you that your manuscript has been deemed suitable for publication in PLOS ONE. Congratulations! Your manuscript is now being handed over to our production team.

Kind regards, 

on behalf of

Professor Mukhtiar Baig 

Academic Editor

PLOS ONE